# Copper intrauterine device increases vaginal concentrations of inflammatory anaerobes and depletes lactobacilli compared to hormonal options in a randomized trial

Bryan P. Brown [1,2] ✉, Colin Feng[1], Ramla F. Tanko[3,4], Shameem Z. Jaumdally[3], Rubina Bunjun [3], Smritee Dabee [1,2,3], Anna-Ursula Happel[3], Melanie Gasper[1,2], Donald D. Nyangahu[1,2], Maricianah Onono[5], Gonasagrie Nair[6], Thesla Palanee-Phillips[7], Caitlin W. Scoville[2], Kate Heller[2], Jared M. Baeten [2,8], Steven E. Bosinger [9,10], Adam Burgener[11,12,13], Jo-Ann S. Passmore [3,14], Renee Heffron[2,15] & Heather B. Jaspan[1,2,3] ✉

Effective contraceptives are a global health imperative for reproductive-aged women. However, there remains a lack of rigorous data regarding the effects of contraceptive options on vaginal bacteria and inflammation. Among 218 women enrolled into a substudy of the ECHO Trial (NCT02550067), we evaluate the effect of injectable intramuscular depot medroxyprogesterone acetate (DMPA-IM), levonorgestrel implant (LNG), and a copper intrauterine device (Cu-IUD) on the vaginal environment after one and six consecutive months of use, using 16S rRNA gene sequencing and multiplex cytokine assays. Primary endpoints include incident BV occurrence, bacterial diversity, and bacterial and cytokine concentrations. Secondary endpoints are bacterial and cytokine concentrations associated with later HIV seroconversion. Participants randomized to Cu-IUD exhibit elevated bacterial diversity, increased cytokine concentrations, and decreased relative abundance of lactobacilli after one and six months of use, relative to enrollment and other contraceptive options. Total bacterial loads of women using Cu-IUD increase 5.5 fold after six months, predominantly driven by increases in the concentrations of several inflammatory anaerobes. Furthermore, growth of *L. crispatus* (MV-1A-US) is inhibited by $Cu^{2+}$ ions below biologically relevant concentrations, in vitro. Our work illustrates deleterious effects on the vaginal environment induced by Cu-IUD initiation, which may adversely impact sexual and reproductive health.

Unintended pregnancy can have grave consequences for women, children, and families, making access to safe and effective contraception a global health priority[1,2]. Modern contraceptive methods are used globally, though women living in regions with a high burden of

sexually transmitted infections (STI) face uncertainty concerning contraceptive-induced effects on the vaginal microbiota[3]. Previous studies have suggested that some contraceptives, including both non-hormonal and hormonal options, may alter vaginal microbiota[4–7] and

**Table 1 | Baseline cohort characteristics**

| | Copper IUD (N = 62) | DMPA-IM (N = 64) | LNG Implant (N = 68) |
|---|---|---|---|
| Age[a] | 24.0 (21.0, 27.0) | 24.0 (21.0, 26.3) | 23.0 (21.0, 26.0) |
| BMI[a] | 25.1 (21.6, 30.7) | 24.5 (22.2, 30.3) | 23.0 (21.0, 30.6) |
| Gravidity[a] | 1 (1, 2) | 1.00 (1, 2) | 1.00 (1, 2) |
| Term pregnancies[a] | 1 (1, 2) | 1.00 (1, 2) | 1.00 (1, 2) |
| Primary partners[a] | 1 (1, 1) | 1.00 (1, 1) | 1.00 (1, 1) |
| **STI prevalence** | | | |
| *Chlamydia trachomatis*[b] | 14 (23%) | 9 (14%) | 5 (7%) |
| *Neisseria gonorrhoea*[b] | 5 (8%) | 3 (5%) | 3 (4%) |
| HSV-2[b] | 25 (40%) | 34 (53%) | 30 (44%) |
| **Clinical BV prevalence** | | | |
| BV positive[b] (Nugent 7–10) | 20 (34%) | 15 (26%) | 21 (34%) |
| BV intermediate[b] (Nugent 4–6) | 9 (15%) | 6 (10%) | 5 (8%) |
| BV negative[b] (Nugent 0–3) | 30 (51%) | 37 (64%) | 35 (57%) |
| **CST distribution** | | | |
| I-B[b] | 7 (11%) | 6 (9%) | 4 (6%) |
| III-A[b] | 23 (37%) | 33 (52%) | 26 (38%) |
| IV-A[b] | 12 (19%) | 9 (14%) | 10 (15%) |
| IV-B[b] | 9 (15%) | 5 (8%) | 13 (19%) |
| IV-D[b] | 11 (18%) | 11 (17%) | 15 (22%) |

[a]Median (IQR).
[b]n (%).

host mucosal immune responses[8–11], which may influence adverse health outcomes in women. The composition of vaginal microbiota has been identified as a significant determinant of reproductive health[12–14]. Several clinical studies have shown that bacterial vaginosis (BV), characterized by a diverse community of anaerobic bacteria and depletion of *Lactobacillus* spp., is associated with elevated STI risk compared to communities dominated by lactobacilli (especially non-*iners Lactobacillus* spp.)[14,15]. In agreement, lactobacilli-dominated vaginal communities are an important component of mucosal defense against STIs[16]. In addition to lowering vaginal pH as a result of their metabolites, there are several modes of action by which beneficial vaginal bacteria reduce STI susceptibility, including modulation of epithelial barrier integrity[17]; alteration of epithelial or immune cell function via production of cytokines or other factors[8,18–20].

The Evidence for Contraceptive Options and HIV Outcomes (ECHO) Trial found no significant difference in HIV incidence rates among 7,829 African women randomized to either (a) injectable intramuscular depot medroxyprogesterone acetate (DMPA-IM), (b) the levonorgestrel (LNG) implant, or(c) the non-hormonal T-380 copper intrauterine device (Cu-IUD)[21]. Following these findings, the World Health Organization (WHO) modified their recommendations that women can use any method of contraception, barring medical contraindications, regardless of their HIV risk[22]; however, these recommendations do not take into account possible interactions between contraceptives and resident microbiota of the female genital tract (FGT). Data conflict as to if and how hormonal and non-hormonal contraceptives modulate vaginal microbiota[23,24]. Injectable progestin use has been implicated in increased risk for STIs[18,23,25] but the underlying mechanisms remain unclear[26–29]. Specifically, DMPA-IM administration has been associated with a reduction in vaginal epithelial layer thickness[6], which in turn may impair colonization of lactobacilli[6,23]. However, in some observational studies, DMPA-IM has been associated

with decreased risk of BV[30–32], while Cu-IUD initiation has been found to increase BV prevalence[4,33]. BV itself is associated with increased STI incidence, including *Chlamydia trachomatis* and *Neisseria gonorrhoeae*[34], as well as with increased risk for other adverse reproductive health outcomes including pelvic inflammatory disease[25], and a myriad of adverse birth outcomes[35–39].

To date, the association between initiation of three commonly used contraceptives, DMPA-IM, LNG implant, and the Cu-IUD, and concomitant changes in the vaginal environment has not been explored in a randomized trial, which overcomes biases introduced by differences in behavior among women choosing their preferred contraception in observational studies. The recently completed UChoose Trial found differential effects of three contraceptives on the FGT microbiota and cytokine profile[3], though all of those options were hormonal and distinct from those used in this study. Here, we present data on shifts in vaginal bacterial community dynamics, total bacterial loads and the inferred concentrations of specific taxa, clinical BV prevalence, and subsequent effects on the cytokine milieu in genital samples collected prospectively from a subset of South African and Kenyan women randomized to DMPA-IM, LNG implant, and Cu-IUD within the ECHO Trial. We experimentally interrogated possible mechanisms for our findings in vitro, identifying ionic copper-induced growth inhibition of *L. crispatus*. We further perform an integrative analysis of microbial and inflammatory profiles in participants who acquired HIV versus those who remained seronegative during the trial, identifying microbial and inflammatory factors delineating these groups.

## Results

### Enrollment characteristics

Samples from 218 women enrolled in the nested mucosal sub-study of the ECHO Trial (Figure S1), from South Africa and Kenya, were included in this analysis. The median age of the women was 24 years (IQR 21–27) and participants had a median of one current partner (IQR 1–1) (Table 1). At enrollment, there were no significant differences in distributions of clinical BV status, *Neisseria gonorrhoeae*, or HSV-2 serostatus between arms (Table 1). The prevalence of *Chlamydia trachomatis* differed by contraceptive arm, with 23% prevalence among women assigned to Cu-IUD, 14% in women assigned to DMPA-IM, and 7% in women assigned to LNG implant (Table 1), though this did not have a significant effect on bacterial community alpha- or beta-diversity (Figure S2).

### Vaginal bacterial communities

After applying sequencing quality filtering and depth cutoffs, a total of 542 lateral vaginal wall samples from 218 women from a median of three study visits were used in this analysis. The 106 missing samples from these women were evenly distributed across visits and arms and were not significantly associated with either variable ($\chi^2 P = 0.88$). PAM clustering of community profiles using Bray-Curtis distance on relative abundance transformed taxonomic profiles resulted in the identification of five community state types (CSTs; Figure S3). CSTs III-A and I-B were numerically dominated by *Lactobacillus* taxa (*L. iners* and *L. crispatus*, respectively) and displayed low Shannon diversity. CST IV-B was primarily comprised of *Gardnerella vaginalis* and displayed a moderate level of diversity. CSTs IV-A and IV-D were strongly correlated with clinical BV diagnosis and displayed the highest Shannon diversity, with *Lachnocurva vaginae* as the most abundant taxon in CST IV-A (Figure S3).

### Copper IUD significantly elevates vaginal bacterial alpha diversity and Nugent score

After adjusting for baseline Nugent score and multiple comparisons, clinical BV classification (as measured via Nugent score) was significantly higher among participants randomized to Cu-IUD after

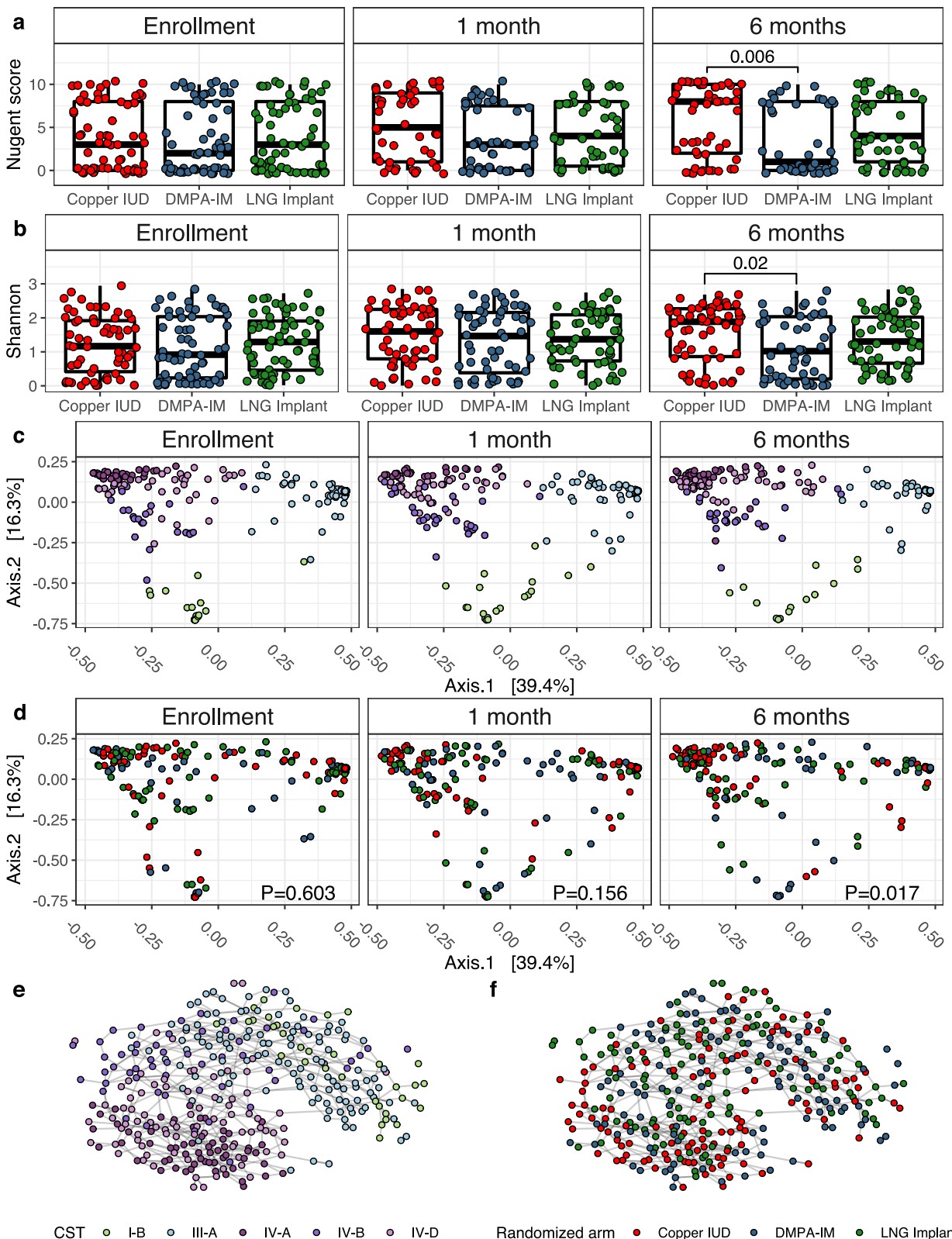

6 months of use (ANCOVA adjusted $P = 0.0006$; Fig. 1a; enrollment median: 3 and IQR: 0–8; 6-month median: 8, and IQR: 2–10) relative to participants randomized to DMPA-IM. After 6 months of contraceptive use, Nugent scores were significantly higher (Wilcoxon $P = 0.026$) than the enrollment visit, though only for participants randomized to Cu-IUD (Figure S4A). Likewise, women randomized to Cu-IUD displayed significantly higher bacterial Shannon diversity compared to

participants randomized to DMPA-IM (ANCOVA adjusted $P = 0.02$ after adjusting for baseline Shannon and multiple comparisons; Fig. 1b), although this was not evident after 1 month (Fig. 1b). At the 6-month visit, women randomized to DMPA-IM also had lower Shannon diversity than women randomized to LNG implant, although this did not reach statistical significance (ANCOVA adjusted $P = 0.16$; Fig. 1c). Among Cu-IUD users, alpha diversity increased, though not

**Fig. 1 | Contraceptive use significantly alters the composition and diversity of vaginal bacterial communities across 218 participants. a** Cross-sectional comparisons of clinical Nugent scores by randomization arm Cu-IUD ($n = 47$); DMPA-IM ($n = 36$); LNG implant ($n = 46$). **b** Cross-sectional comparison of the effect of each contraceptive method on vaginal bacterial Shannon diversity Cu-IUD ($n = 55$); DMPA-IM ($n = 45$); LNG implant ($n = 53$). **c** PCoA ordination of relative abundance transformed taxonomic abundance using Bray-Curtis distance. Samples are colored by CST. **d** PCoA ordination of relative abundance transformed taxonomic abundance using Bray-Curtis distance. Samples are colored by randomization arm.

**e** Network analysis of samples visualized using a Bayesian directed acyclic graph (DAG) and colored by CST. **f** DAG colored by randomization arm. DAGs were generated using Euclidean distance on centered log ratio transformed taxonomic abundance. Network inference was performed using the PC algorithm. Two-tailed $P$ values were calculated using an ANCOVA model for multiple comparisons of Nugent scores and Shannon diversity, and a permutational ANOVA for multiple comparisons in the beta-diversity analysis. All hypothesis tests included baselines values as a covariate. Boxplots limits define the first and third quartiles, the center line represents the median, and whiskers define 1.5x interquartile range.

significantly, from enrollment to 1 month (Shannon index median at enrollment and 1 month: 1.17 and 1.6, respectively; Wilcoxon $P = 0.062$; Figure S4B), which was not evident in women using DMPA-IM or LNG implant. There was no significant difference in Shannon diversity by randomization arm at enrollment (Figure S4B). Cu-IUD use resulted in a significant increase in Shannon diversity after 6 months of use (Shannon index median: 1.89; Wilcoxon $P = 0.017$; Figure S4B) compared to enrollment that was not observed in women randomized to the other methods. There was also no significant difference in bacterial alpha diversity between participants who tested positive for *C. trachomatis* or *N. gonorrhoeae* at enrollment versus those who tested negative (Shannon: Wilcoxon $P = 0.36$; Figure S2A, B).

## Contraceptive use causes shifts in bacterial community profiles

To assess the effect of contraceptive use or a covariate on bacterial community profiles, we compared inter-community (between sample) distance within categorical groupings (e.g. randomized arm, study site, STI status etc.) against the across-group centroid using a PERMANOVA. We found no significant clustering of samples by enrollment randomization arm using Bray-Curtis distance (PERMANOVA $P = 0.603$; PERMDISPERSION $P = 0.385$). Similarly, *C. trachomatis* and *N. gonorrhoeae* infections at enrollment did not have a significant effect on inter-community distance (Bray-Curtis; PERMANOVA $P = 0.121$, PERMDISPERSION $P = 0.927$; Figure S2C, D). However, because STI status was imbalanced between groups at enrollment, these were adjusted for in downstream analyses. Study site did have a statistically significant effect on inter-community distance at enrollment (Figure S2E; Bray-Curtis; PERMANOVA $P = 0.024$; PERMDISPERSION $P = 0.53$) and 1 month of contraceptive use (Bray-Curtis; PERMANOVA $P = 0.028$; PERMDISPERSION $P = 0.42$), but not at 6 months (Bray-Curtis; PERMANOVA $P = 0.072$; PERMDISPERSION $P = 0.19$) and therefore was also included as a covariate in differential abundance testing and regression models evaluating bacterial abundance. Randomized contraceptive arm had a significant effect on inter-community distance after 1 month of use (PERMANOVA $P = 0.024$; PERMDISPERSION $P = 0.054$; Fig. 1d), though not when controlling for study site, STI status at enrollment, and enrollment CST (PERMANOVA $P = 0.156$). Contraceptive method was a significant determinant of inter-community distance after 6 months of use (PERMANOVA $P = 0.006$; PERMDISPERSION $P = 0.759$) and remained so when controlling for study site and enrollment CST (PERMANOVA $P = 0.017$).

We generated Bayesian directed acyclic graphs (DAGs), which are useful for inferring relationships between microbial communities[40], to infer the relationships between samples, CSTs, and the effect of contraceptive use. When generating DAGs from participant samples after 1 and 6 months of contraceptive use, participants randomized to Cu-IUD clustered consistently in the same Euclidean space as those with diverse, BV-associated CSTs (IV-A and IV-D; Fig. 1e, f). Conversely, participants randomized to LNG implant and DMPA-IM clustered in spaces associated with low diversity CSTs. Cu-IUD use was significantly associated with a shift to non-*Lactobacillus*-dominant bacterial communities after 6 months of use, as compared to both hormonal options (Boschloo's $P = 0.03$, OR $= 1.85$).

## Distribution of community state types and transition frequencies induced by contraception initiation

Among the five CSTs identified, there was no significant difference in the distribution of these CSTs among randomization arms at enrollment (Fig. 2a). When evaluating the effects of contraceptive use on the distribution of CSTs, we observed differential shifts through time (Fig. 2) by randomization arm. While none of the five detected CSTs were significantly associated with users of any contraceptive after 1 month, CST IV-A was significantly more prevalent in participants randomized to Cu-IUD at 6 months (11.6%) than either of the other contraceptives (2.9% of DMPA-IM and 4.6% of LNG-implant, McNemar's $\chi^2$ $P = 0.003$). CST IV-B was significantly more prevalent among participants using LNG implant (8.1%) for 6 months than any other contraceptive (2.3% for DMPA-IM and 3.5% for Cu-IUD, $\chi^2$ $P = 0.03$). We also report a significantly lower prevalence of CST I-B in women randomized to Cu-IUD after 6 months compared to DMPA-IM for the same duration (Boschloo's $P = 0.016$, OR $= 0.184$), though not relative to LNG implant (Boschloo's $P = 0.19$, OR $= 0.43$). Relative to baseline, the prevalence of CST I-B decreased 3.5-fold after 6 months of Cu-IUD use, but when subset to paired samples this did not remain significant (McNemar's $\chi^2$ $P = 0.1$). When evaluating transitions between CSTs, we found that participants randomized to DMPA-IM showed a statistically significant increase in the frequency of transition from CST III-A to I-B ($\chi^2$ $P = 0.008$) after 6 months of use.

## Changes in bacterial taxa induced by contraceptive initiation

We used ANCOM-BC and DESeq2 to identify taxa that were significantly differentially abundant at 1 and 6 months post-contraception initiation (Fig. 2b and Figure S5) compared to enrollment. In addition to the initial prevalence and relative abundance filters, we required that a taxon be present across 15% of samples to be included in differential abundance testing. Participants randomized to Cu-IUD displayed significant reductions in *Lactobacillus* species as well as increases in several BV-associated bacteria by both methods after 1 and 6 months of use. Cu-IUD was the only contraceptive that was associated with significant alterations in bacterial relative abundance after 1 month of use, resulting in increases in *Sneathia amnii* and *Veillonella montpellierensis* by both methods and increases in additional dysbiotic taxa along with a decrease in *L. iners* as detected by ANCOM-BC (Fig. 2b). After 6 months, participants randomized to Cu-IUD displayed significant reductions in the abundance of *Limosilactobacillus reuteri* (formerly *Lactobacillus reuteri*; DESeq2; Figure S5) and *Corynebacterium amycolatum* (ANCOM-BC), and elevations in the abundance of *S. amnii* and *Prevotella timonensis (both methods)* as well as other BV-associated taxa by one or both methods. Participants randomized to DMPA-IM displayed significant increases in the relative abundance of *S. amnii* (both methods) after 6 months of use. Participants randomized to LNG implant experienced a small but significant increase in the abundance of a single *Sneathia* taxon (DESeq2) at 6 months. All of these shifts were significant after correction for multiple comparisons via the Benjamini-Hochberg method[41].

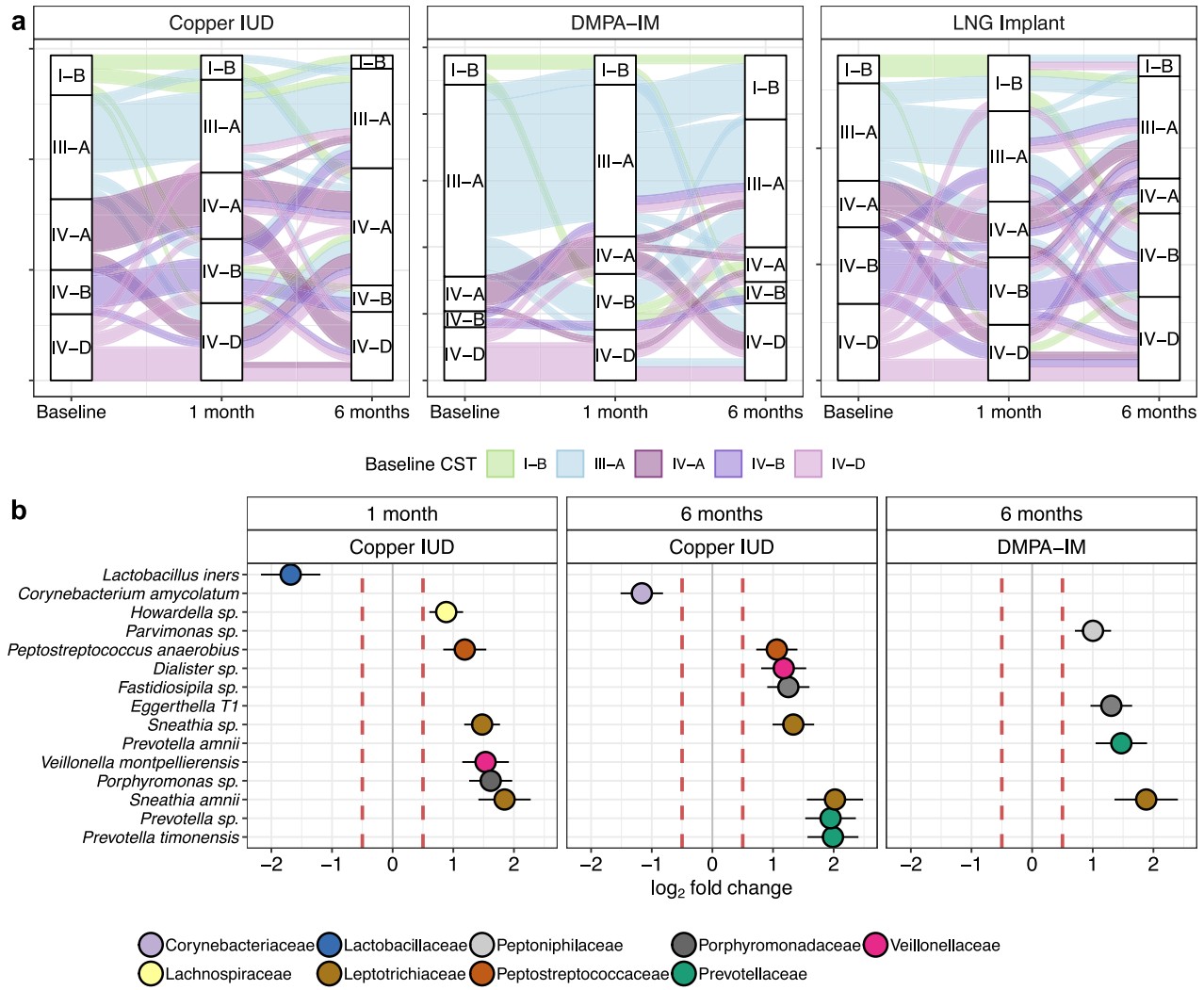

**Fig. 2 | Contraceptive use drives CST transition and shifts in bacterial abundance. a** CSTs are indicated at each time point and alluvials track the progression of a given participant across time. Plots are faceted by randomization arm. **b** Fold changes in bacterial abundance after 1 and 6 months of contraceptive use, relative to enrollment abundance. Fold changes are log₂ transformed and were calculated using the default (two-sided) ANCOM-BC model for two-group comparisons. Vertical dashed lines indicate a 0.5-fold change. Each point represents the arithmetic mean log₂ fold change and solid horizontal lines represent the standard error. Taxa with P < 0.05 after adjustment for multiple comparisons (Benjamini and Hochberg) are shown; Cu-IUD (n = 55); DMPA-IM (n = 45); LNG implant (n = 53).

### Cu-IUD use results in significant increases in total bacterial load and abundance of several BV-associated taxa

Total bacterial loads and the inferred concentrations of several taxa were significantly altered by contraceptive use (Fig. 3). After 6 months of use, women randomized to Cu-IUD displayed a 3.4-fold increase in the median bacterial load (Student's P = 8.2e−4, enrollment median: 12,000, and IQR: 3655–42,240; 6 month median: 40,501, and IQR: 16,867–124,630). To a lesser extent, women randomized to DMPA-IM for 6 months also displayed a significant increase in total bacterial load (Student's P = 0.019, enrollment median: 10,257, and IQR: 1915–34,303; 6 month median: 36,309, and IQR: 6795–99,135), as well as women randomized to LNG implant (Student's P = 0.026, enrollment median: 8175, and IQR: 2414–35,472; 6 month median: 20,113, and IQR: 3758–129,275) (Fig. 3a). Although the use of all three options resulted in significant elevations in total bacterial load after Benjamini-Hochberg correction[41], both the magnitude of increase and the taxa driving these shifts were distinct. After adjustment for multiple comparisons, Cu-IUD use was uniquely associated with an increase in the absolute abundance of several dysbiotic, proinflammatory, and/or BV-associated taxa, such as: *A. vaginae* (Wilcoxon P = 0.032), *Dialister* sp.

(Wilcoxon P = 0.025), *Eggerthella Type I* (Wilcoxon P = 0.03), *Fastidiosipila* sp. (Wilcoxon P = 0.03), *G. vaginalis* (Wilcoxon P = 0.03), *Peptostreptococcus* sp. (Wilcoxon P = 0.018), *P. timonensis* (Wilcoxon P = 0.007), *Prevotella* sp. (Wilcoxon P = 0.025), *S. amnii* (Wilcoxon P = 0.01), and *Sneathia* sp. (Wilcoxon P = 0.0; Fig. 3b). Conversely, women randomized to DMPA-IM exhibited increased concentrations of *L. iners*, which mostly drove the increase in total bacterial load associated with this contraceptive, though the increase did not remain significant after correction (Figure S6A). None of these contraceptives significantly altered the absolute abundance of *L. crispatus*, but this is likely due to its low prevalence across the dataset (Figure S6B). All reported *p*-values associated with differentially abundant taxa were corrected using the method of Benjamini and Yekutieli[42].

### Nugent score increases with Cu-IUD use and strongly correlates with Cu-IUD induced alterations in absolute bacterial abundance

To relate individual bacterial concentrations to broader health metrics such as clinical BV diagnosis, we calculated the sum of the concentrations of bacteria that were significantly altered by each

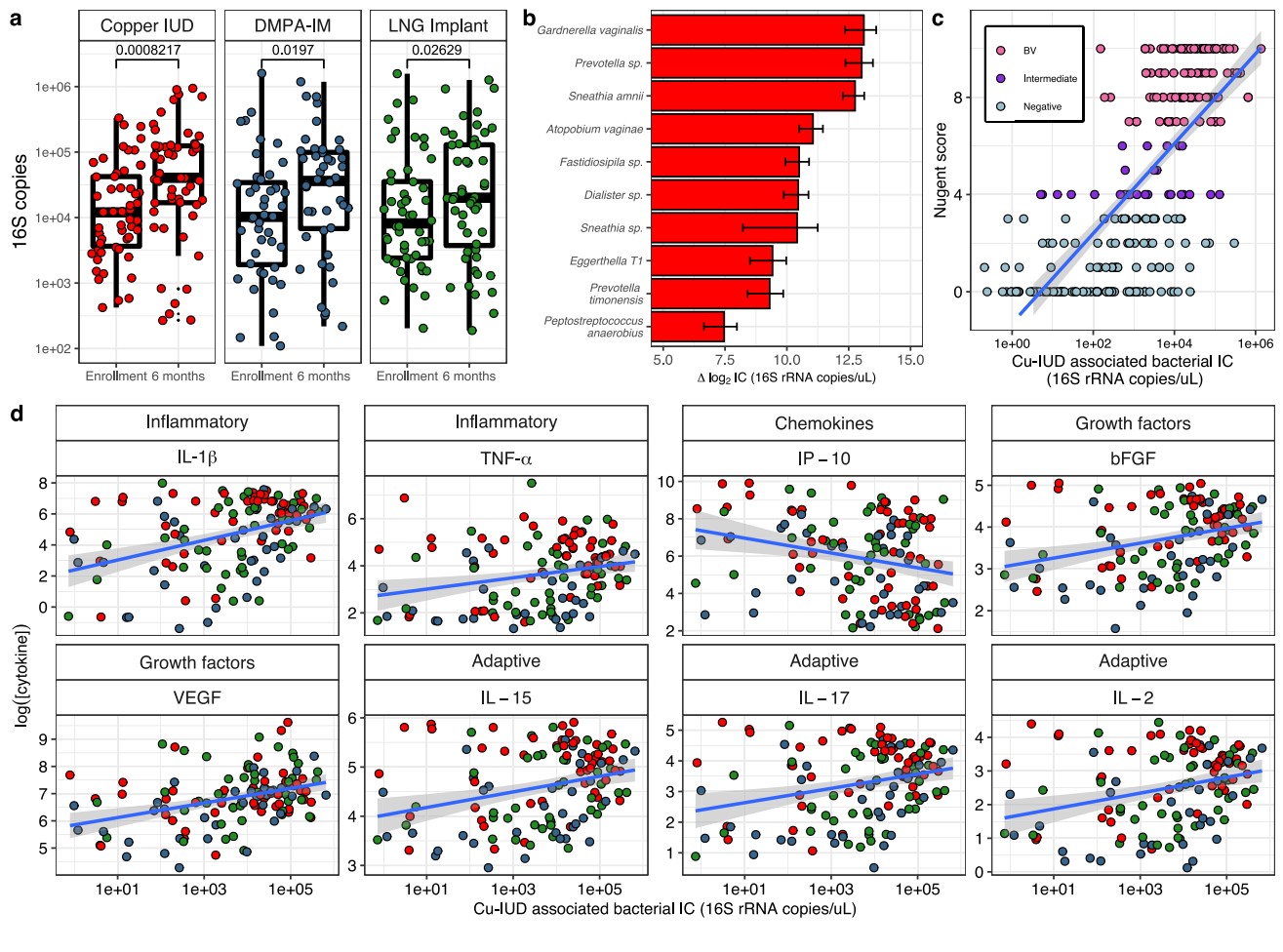

**Fig. 3 | Cu-IUD use induces significant increases in the concentrations of several cervicovaginal bacteria and cytokines. a** Absolute 16S rRNA gene copy number assessed by broad-range qPCR at the study enrollment visit and after 6 months of the assigned contraceptive use. *P* values were calculated using a Student's t test and corrected using the Benjamini and Hochberg method; Cu-IUD (*n* = 54); DMPA-IM (*n* = 44); LNG implant (*n* = 52). **b** Increases in absolute bacterial abundance after 6 months of Cu-IUD use. Only those taxa whose abundance remained significant after adjustment for multiple comparisons (Benjamini and Yekutieli) are shown. Data are presented as mean values +/− SEM. **c** The additive concentration of bacteria altered by Cu-IUD use correlates strongly with Nugent score. **d** The additive concentration of bacteria altered by Cu-IUD use correlates strongly with the concentration of several key cytokines after 6 months of contraceptive use. Two-tailed *P* values were derived from the t-value of the generalized linear model. Cytokine and bacterial concentrations were log$_2$ transformed prior to analysis. The class of cytokine is indicated above each subplot. Participant data points are colored by randomization arm and are from data collected after 6 months of contraceptive use. Boxplot center lines indicate the median, while the hinges indicate the first and third quartiles, and whiskers extend to 1.5 * IQR from the given hinge. Models with *p* < 0.05 after adjustment for multiple comparisons (Benjamini and Yekutieli) are shown. Regression lines are indicated in blue and the shaded regions around regression lines represent the 95% confidence interval.

contraceptive. DMPA-IM and LNG implant use did not result in significant shifts in the concentrations of any bacteria after adjustment for multiple comparisons and thus were not correlated with Nugent score. Conversely, the concentrations of bacteria whose abundance was altered after 6 months of Cu-IUD use (Fig. 3b), was a significant predictor of clinical BV diagnosis via Nugent score (t (271 degrees of freedom) = 17.217, *P* < 2e−16; Fig. 3c).

**Shifts in bacterial abundance induced by Cu-IUD initiation are tightly correlated with the abundance of several key cytokines**
We measured the concentrations of 27 cytokines in cervicovaginal fluid (Tanko et al.) and assessed the relationship between vaginal microbiota and cytokine concentrations. To link shifts in the bacterial microbiota to alterations in cytokine concentration, we regressed the sum of the inferred concentrations of taxa significantly altered by Cu-IUD use after 6 months of use against the concentrations of the 26 measured cytokines that passed our

quality controls. In addition to correlating strongly with Nugent score (Fig. 3c), the concentrations of Cu-IUD altered bacteria was highly predictive of the concentrations of 8 of the 26 cytokines measured in menstrual cup cervicovaginal fluid after correction for multiple comparisons. After 6 months of contraceptive use, the concentrations of Cu-IUD altered bacteria were positively correlated with the concentrations of inflammatory (IL-1β and TNF-α) and adaptive cytokines (IL-2, IL-15, IL-17), several growth factors (FGF-basic, VEGF), and the chemokine IP-10 (Fig. 3d, Table S2). As no bacterial concentrations were significantly altered after 6 months of LNG Implant or DMPA-IM use, these options were *de facto* not correlated with the concentrations of any cytokines. We also observed that several cytokines were strongly positively correlated with Shannon entropy measures (Figure S7), though prediction accuracy (Table S2) was lower than for models using the inferred concentrations of Cu-IUD altered bacteria as the predictor variable.

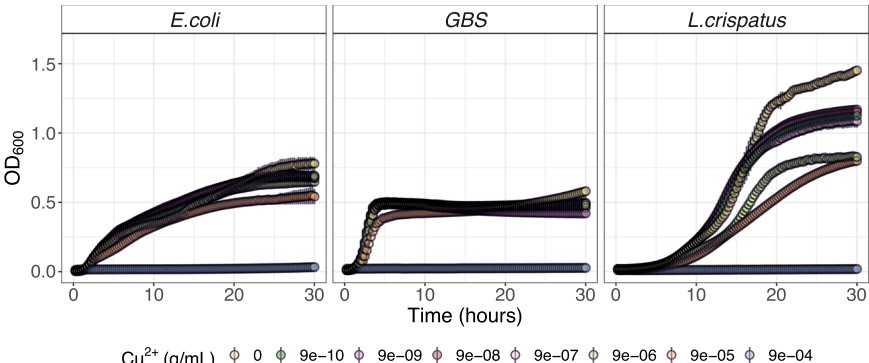

**Fig. 4 | Growth curves of *Lactobacillus crispatus*, *Streptococccus agalactiae*, and *Escherichia coli* over 30 h in the presence of media alone (yellow) or increasing concentration of Copper (Cu²⁺) ions.** Data represent triplicates of each sample at each Cu²⁺ dilution and are represented as the arithmetic mean (point) and standard error (vertical line). Experiments were performed three times.

## Copper ions inhibit the growth of *L. crispatus* across a range of biologically relevant concentrations

We measured the growth of *L. crispatus* (MV-1A-US, BEI Resources), *Streptococcus agalactiae* (Group B *Streptococcus COH1*; GBS COH1), and *E. coli* (DH10B, Thermo Fisher) in vitro in the presence of ionic copper at concentrations spanning several orders of magnitude ($9e^{-4}$: $9e^{-10}$ g/mL) similar to those measured in cervical secretions during Cu-IUD use[43]. Among the three bacteria assayed in triplicate, only *L. crispatus* displayed reduced growth across multiple copper concentrations examined. While the growth of all three taxa was completely inhibited by ionic copper (Cu²⁺) concentrations of $9e^{-4}$ g/mL, *L. crispatus* growth was also severely reduced in a dose-dependent manner, with substantial inhibition at $9e^{-5}$ and $9e^{-6}$ g/mL (Fig. 4). Our data also revealed that even ionic copper concentrations as low as $9e^{-10}$ g/mL were sufficient to reduce the growth of *L. crispatus* by an average of 23% after 30 h of growth, which was not observed for other taxa.

## Integrative compositional analyses reveal alterations in microbiota and inflammation between cases and controls

We compared the vaginal microbiota of women who acquired HIV during follow-up (cases) and age and site matched controls using samples collected at the visit prior to seroconversion (or equivalent time point for controls). Cases were fairly evenly distributed across all five CSTs, except no seroconversion occurred among women with CST I-B, which is numerically dominated by *L. crispatus* (Table 2; Figure S8A). We used a sparse principal component (SPC) analysis to identify which individual bacterial concentrations or cytokines best distinguished cases and controls. When decomposing our integrated dataset, we found that the fourth component explained the variance that was most associated with, and significantly segregated, case samples from controls (Fig. 5a). The abundance of bacterial taxa loaded SPC4 more strongly than cytokine concentrations, with the absolute abundance of *L. crispatus* being the factor that had the greatest contribution. In addition, *M. hominis* and *G. asaccharolytica* loaded SPC4 strongly, as well as cytokines IL-1B, IL-6, MIP-1a, and IP-10, among others. We additionally used ANCOM-BC and DESeq2 to perform differential abundance testing between cases and controls. We found that cases had greater than 10 log₂ fold relative abundance of

*TM7-H1* and *Prevotella ihumii* compared to control participants (DESeq2; Figure S8B). No taxa were detected as significantly differentially abundant by ANCOM-BC after adjustment, which we expect was due to the small number of cases. As with the pre-post analysis, all p-values were adjusted using the method of Benjamini and Hochberg[41].

## Discussion

Our substudy, which included South African and Kenyan women enrolled in the ECHO Trial[21], assayed the effects of the Cu-IUD, DMPA-IM, and LNG implant on the vaginal microbiota at enrollment and after 1 and 6 months of use. We found that women initiating Cu-IUD experienced an increase in vaginal bacterial diversity after 1 month of use, with a more pronounced increase after 6 months of use. Cu-IUD initiation also resulted in significantly higher vaginal bacterial diversity compared to DMPA-IM after 6 months of use when controlling for baseline values. Conversely, initiation of DMPA-IM or LNG implant was not associated with significant cross-sectional or longitudinal shifts in vaginal bacterial diversity. The effects of Cu-IUD use similarly extend to clinical BV status (assayed via Nugent scoring), where we found that women randomized to Cu-IUD for 6 months displayed significantly higher Nugent scores than women using DMPA-IM. Using broad-range qPCR, we found that both Cu-IUD and DMPA-IM use resulted in significant increases in the total vaginal bacterial load after 6 months of use, though this increase was more pronounced, and driven by increases in the concentrations of dysbiotic taxa, for Cu-IUD users. The increase in bacterial load driven by DMPA-IM use appeared to be driven by an increase in the concentration of *L. iners*. Furthermore, after 6 months of use, Cu-IUD associated increases in BV-associated bacterial concentrations were significantly correlated with the concentrations of several cervicovaginal cytokines, which has broad, negative implications for genital health and STI susceptibility[44]. More broadly, contraceptive use significantly affected bacterial community composition after 1- and 6-months of use. Participants using Cu-IUD largely clustered in the same regions as BV-associated CSTs in decomposed space. Conversely, women randomized to the LNG implant and DMPA-IM more consistently clustered in regions associated with mid or lower-diversity CSTs. Furthermore, women randomized to Cu-IUD displayed significant elevations in the relative abundance of bacterial taxa previously linked to BV or other negative health outcomes[45], with Cu-IUD initiation also leading to a 1.68 log₂ fold reduction in *L. iners* relative abundance compared to pre-initiation levels. Collectively, our results argue that Cu-IUD elicited the most robust effects on the vaginal environment, while DMPA-IM and the LNG implant were associated with relatively minor, if any, shifts in bacterial community composition or inflammation. If anything, DMPA-IM was beneficial for the vaginal environment, resulting in significant shifts toward an *L. crispatus* dominant CST and in increase in the

**Table 2 | Counts and percentages of cases and controls by vaginal community state type**

| CST | I-B | III-A | IV-A | IV-B | IV-D |
|---|---|---|---|---|---|
| Case, *n* (%) | 0 (0%) | 9 (39%) | 2 (9%) | 1 (4%) | 11 (48%) |
| Control, *n* (%) | 10 (11%) | 33 (35%) | 17 (18%) | 9 (9%) | 26 (27%) |

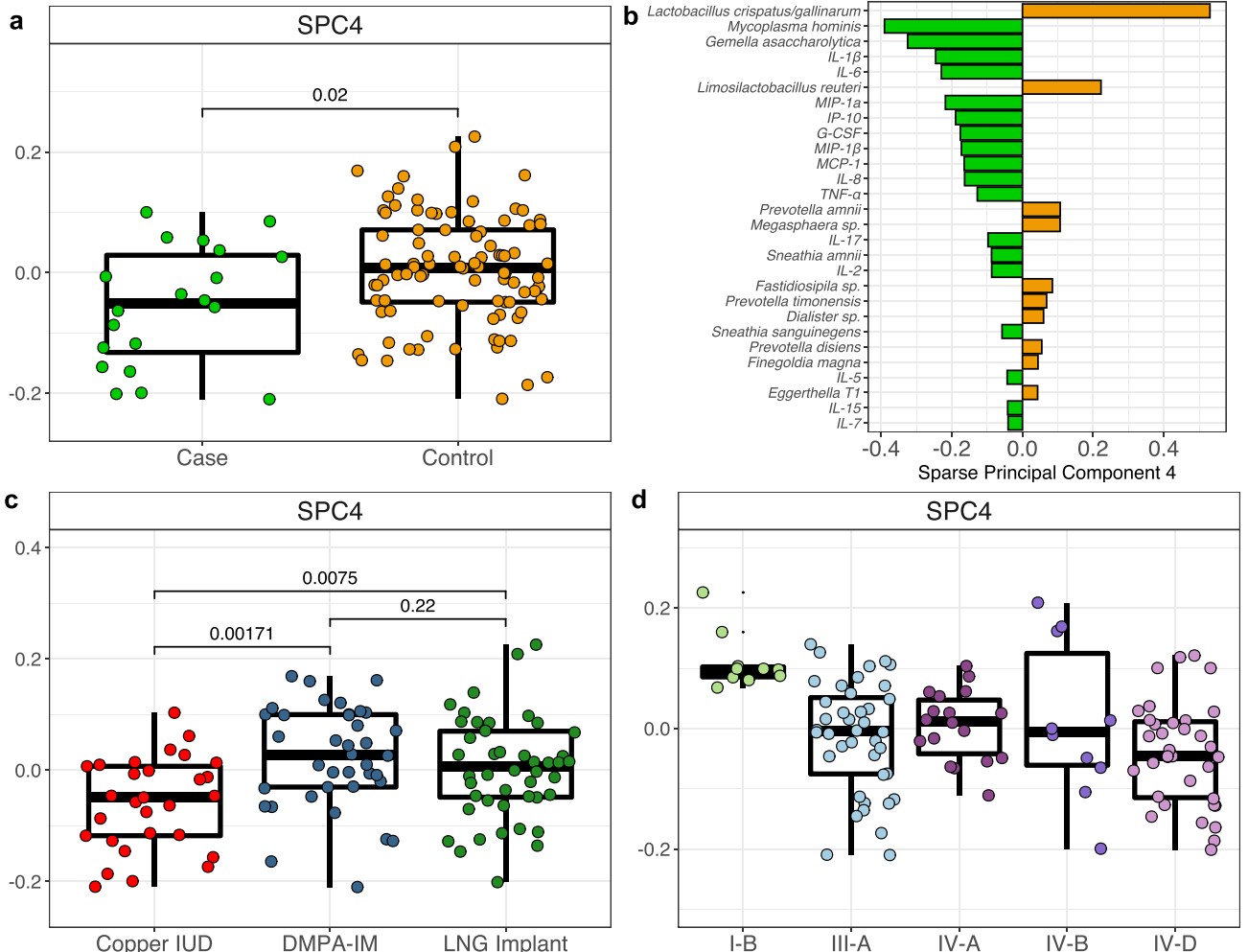

**Fig. 5 | Integrative analysis of bacterial taxa and cytokines differentiating cases** (*n* = 22) **from matched controls** (*n* = 94) **in the substudy. a** Boxplots displaying the distribution of scores along the fourth sparse principal component (SPC4), which was distinct for cases (*n* = 20) and controls (*n* = 93). **b** Loadings scores for SPC4, which segregates cases from controls. Bars are colored according to which group had higher median scores for SPC4. **c** The distribution of SPC4 scores

grouped by contraceptive arm Cu-IUD (*n* = 29); DMPA-IM (*n* = 38); LNG implant (*n* = 46). **d** The distribution of SPC4 scores grouped by CST (*n* = 113 participants). Boxplots display the first and third quartiles and the median value. Two-tailed *P* values were calculated with a Wilcoxon Rank Sum test and corrected using the method of Benjamini and Hochberg. Bacterial and cytokine absolute concentrations were log$_2$ transformed and mean-centered prior to analysis.

concentration of *L. iners*. Although our understanding of the role of *L. iners* in the vaginal ecosystem is evolving, *L. crispatus* has been previously linked to many beneficial outcomes in the FGT such as impeding the growth of inflammatory bacteria and pathogens[16], and reduced vaginal inflammation and protection against STIs[15].Thus, these results illustrate the need for considering the microbiological consequences of contraceptive options, due to their significant effects on reproductive health.

In light of the negative effect of Cu-IUD use on the relative abundance and prevalence of various lactobacilli, we sought to elucidate if this effect was mediated by Cu$^{2+}$ ions released from the IUD or due to other factors (e.g. mechanical or inflammatory). Previous work has demonstrated bactericidal properties of both metallic[46–48] and ionic[49,50] copper, with its lethality thought to be due to the generation of oxidative stress[51], and disruption of protein folding[52], among other factors. Martin and Suarez found that *L. jensenii* growth was inhibited by CuSO$_4$ in vitro[53], though at concentrations that have not been previously reported in the vaginal environment. We chose to evaluate the effect of Cu$^{2+}$ on *L. crispatus* since we found a reduction in the prevalence of the *L. crispatus* dominated CST (I-B) among participants using Cu-IUD and due to its

known benefits in the FGT[15] Our data suggested that although Cu$^{2+}$ concentrations of 9e$^{-4}$ g/mL were fully inhibitory for all bacteria tested, only *L. crispatus* growth was substantially inhibited at concentrations as low as 9e$^{-10}$ g/mL. While the concentration of Cu$^{2+}$ in the vagina during Cu-IUD use is unclear, Arancibia et al. found that uterine washings from patients using the T-380A Cu-IUD (the device used in the ECHO Trial and this substudy) contained 1.1e$^{-5}$ g/mL of total copper across the first year of use[43]. Thus, our data span physiologically relevant ranges and demonstrate that even concentrations which are several orders of magnitude below those found in uterine washings yield substantial inhibition of *L. crispatus* growth, which may account for its decreased abundance and increased incidence of *Lactobacillus*-deficient states in Cu-IUD users. At the same time, we also appreciate that *L. crispatus* was the only facultative anaerobe used in these experiments and all bacteria were grown under aerobic conditions, which may have selectively increased stress on *L. crispatus*. However, the robust and substantial growth of *L. crispatus* without Cu$^{2+}$ present, and the clear dose-dependent response exhibited suggest that these effects were associated with Cu$^{2+}$ concentrations, as opposed to poor growth or excess stress associated with aerobic conditions.

We employed an integrative approach to identify microbial and immunological differences between HIV cases and controls, finding that the absolute abundance of *L. crispatus* was the most distinct factor, with a marked lower abundance in cases relative to controls. Furthermore, CST I-B, whose abundance was primarily comprised of *L. crispatus*, was the only CST from which no participants seroconverted (though representation in the trial was also the lowest). This is in line with previous work detailing reduced inflammation and HIV acquisition risk associated with *L. crispatus* dominated communities[15]. Sparse Principal Component analysis also identified MIP-1α, IP-10, IL-1β, and IL-6 as differential between cases and controls, similar to our previous report from CAPRISA004[32]. We note that in addition to case samples, participants randomized to Cu-IUD and those displaying CST IV-D also had negative values along the fourth sparse principal component, arguing that each of these groups shared similar concentrations of the bacteria and cytokines that were associated with later seroconversion. Given the small sample size of cases in this substudy of the ECHO Trial, these findings should be evaluated across other cohorts.

The implications of our findings regarding highly effective and available contraceptives provide key insights into shifts in the vaginal environment that may follow contraceptive initiation. However, limitations of the study include the duration of longitudinal data, small sample sizes of some CST groups, and the lack of a control/placebo arm in the Trial. While we appreciate the ethical constraints of offering a placebo to participants seeking safe and effective contraception, we also appreciate the limitation that this places on our ability to assess the effect of these contraceptive options. However, despite the lack of a true control group, we feel that the randomized nature of the trial and the inclusion of pre-post analyses have enabled our results to have some generalizability and offer valuable insights. In a move to expand the contraceptives available to women in settings with a high unmet need for contraception, use of a Cu-IUD has many benefits including long-term efficacy (up to 5 years), a non-hormonal mechanism of action, and typically does not require regular clinical visits. However, our randomized data confirm the observations of others[54] that suggest that Cu-IUD may disrupt vaginal health and may put users at risk for vaginal dysbiosis and downstream sequelae. Initiation of Cu-IUD use has been previously associated with increased colonization by non-optimal bacteria in observational studies[4], and our results add novelty by suggesting community-level and inflammatory marker shifts also occur, in addition to increases in total bacterial load. If nothing else, the results from our in vitro experiments demonstrating the significant inhibition of *L. crispatus* growth carries potentially serious implications for its effect on reproductive health. Both vaginal dysbioses[15] and genital inflammation[32] may have important consequences for women with frequent exposure to STIs and other pathogens, including HIV. Further, bacterial vaginosis can have a large impact on women's quality of life and sexual frequency[55]. Why incidence was not significantly different between arms in the ECHO Trial, despite the clear differences in vaginal microbiota and inflammatory changes, is unclear. It is also possible that these changes were simply not large or persistent enough to impact HIV risk, or that our sample was not representative of the full ECHO cohort. None-the-less, contraceptives that induce these kinds of shifts have implications beyond HIV risk. Thus, it is imperative that we understand the myriad effects of contraceptive methods when considering expansion of contraceptive options accessible in global health settings.

## Methods

### Study approval

The protocol was approved by the Human Research Ethics Committee of the University of Washington (STUDY00000261), Kenya Medical Research Institute Scientific and Ethics Review Unit (SERU/CMR/P0014/3109), University of Witwatersrand Human Research Ethics Committee (HREC PRC 141112), University of Cape Town Human Research Ethics Committee (HREC 371/2015), and FHI360 (523201). Women provided written informed consent and remuneration of participants was done in accordance with the requirements of local ethics committees to provide fair compensation without inducement.

### Study cohort

The ECHO Trial (clinicaltrials.gov identifier: NCT02550067) compared the relative HIV-1 incidence among women randomized to Cu-IUD, LNG implant, or DMPA-IM as the primary outcome. This nested mucosal sub-study aimed to evaluate the impact of these contraceptives on genital tract microbiota and immunity. The primary endpoints for this mucosal substudy were changes in Th17 cell frequency, microbial diversity and vaginal cytokine concentrations induced by hormonal contraceptive initiation. Secondary outcomes were changes associated with later HIV seroconversion. Eligibility and randomization procedure for the parent trial is described in detail elsewhere[21], but importantly included women seeking effective contraception and being in the age range of 16–35 years. Notably, all participants reported no contraceptive use during the 6 months preceding enrollment and initiated the assigned contraceptive after the enrollment samples were collected. In this sub-study, which included three of the ECHO Trial sites (Cape Town and Johannesburg, South Africa, and Kisumu, Kenya), we consecutively enrolled all eligible women concurrent to their enrollment in the primary trial, or thereafter if already enrolled (only at Kisumu and Cape Town sites). From those who enrolled in this mucosal sub-study, sample size calculations indicated that 20 participants per randomized arm per site with complete sample sets at all three timepoints were needed, and thus we randomly selected women who met this criterion at the Johannesburg, South Africa and Kisumu, Kenya sites. However, at the Cape Town, South Africa site, cervical cytobrushes from 80 consecutively enrolled women were processed for phenotyping ex vivo at enrollment and 1 month post-contraception initiation for a separate study[56], and we elected to include all of these participants, rather than a random subset. In addition, participants that were included in the secondary (case-control) analysis but also had samples available at enrollment, 1 month, and 6 months, were added to the pre-post analysis group if not already included. The complete randomized sample counts and sample counts passing QC for the pre-post analysis are available in Table S3A. For the case-control analysis, only the visit immediately preceding seroconversion (all of which were after contraception initiation) was used in bacterial community analyses. Age and site matched controls were chosen from visits preceding when the matched case tested positive during the study (Table S3B). Controls were randomly selected at a ratio of 4 controls to 1 case, matched on study site, visit, and age. Although the Trial was not blinded, all laboratory personnel receiving, processing, and assaying the samples were blinded, and unblinding only occurred at the time of statistical analysis, after a statistical analysis concept sheet had already been formulated with blinded data. Samples that were included in both analyses due to the availability of samples for all visits for the primary (pre-post) analysis and which were also included in the secondary (case-control) analysis are listed in Table S3C.

**Sample collection.** For cytokine analysis, women had a disposable menstrual cup inserted for approximately 45 min. Clinicians removed the menstrual cup prior to speculum insertion and immediately placed the cup into a 50-mL sterile Falcon tube, at 4 °C for transport to the laboratory within 4 h of collection. At the laboratory, cervicovaginal secretions collected in the menstrual cups were processed and extracted as previously described[57], and diluted 4:1 in sterile phosphate-buffered saline (PBS) and stored at −80 °C. For genomic DNA (gDNA) extraction, a speculum was inserted and vaginal swabs were collected from the lateral vaginal wall. One swab was placed into a sterile vial with Digene® transport media and stored at −80 °C until

gDNA extraction and another swab was rolled onto a slide and fixed for Nugent scoring. An endocervical swab was used for STI testing.

## Clinical diagnostics

At enrollment, endocervical swabs were screened for *Chlamydia trachomatis* (CT) and *Neisseria gonorrhea* (NG). For CT/NG testing, GeneXpert Instrument Systems platform (Cepheid Inc., USA) with the Abbott Real Time PCR assay (Abbott Molecular, USA) were used at South African sites while the Panther System (Hologic Inc., USA) was used at the Kenyan site. Treatment was provided for curable STIs diagnosed syndromically, according to national guidelines, or for CT/NG diagnosed etiologically at enrollment. HSV-2 serology was performed using HerpeSelect ELISA (Focus Diagnostics, USA) at enrollment, and confirmatory testing was performed via Western Blot at the UW. BV was assessed using Nugent scoring criteria[58] at the National Institute for Communicable Diseases. Nugent scores 7–10 were considered BV + while scores 4–6 were considered intermediate BV, and scores 0–3 considered BV−. Only symptomatic BV was treated as per national guidelines.

## Nucleic acid extraction and 16S rRNA gene amplification and sequencing

Prior to nucleic acid extraction, lateral vaginal wall swabs were thawed on ice and shaken at 100RPM for 3 min to mix transport media/sample suspension. In nearly all cases, gDNA was extracted from 250 μL of Digene solution. However, where there was insufficient volume, 550 μL of sterile PBS was added to the swab in the vial, which was shaken for another three minutes at 100RPM, and gDNA was extracted from 250 μL of shaken PBS. Bacterial gDNA was extracted using the DNeasy Powersoil HTP 96 kit (Qiagen) following the manufacturer's protocol. Positive control DNA was extracted on each plate from pure cultures of *E. coli* DH10B[59] alongside sample DNA for cross contamination filtering and error rate modeling. DNA was amplified using primers designed to span the V3-V4 region of the 16S rRNA gene, following the amplification protocol outlined in Gohl et al.[60] except using 319F/806R primers (Table S1). Amplicon libraries were purified using AMPure® XP beads (Beckman Coulter), pooled in equal mass quantities, and quantitated via qPCR using the NEBNext® Library Quant Kit for Illumina (New England BioLabs). Paired-end sequencing was performed on an Illumina Miseq platform using V3 600 cycle kits as previously described[61].

## Cytokine analysis

The concentrations of 27 cytokines were measured by Luminex in cervicovaginal exudates using the Bio-Plex Pro Human 27-plex cytokine assay (Bio-Rad). These included interleukin (IL)-1RA, IL-1β, IL-2, -4, -5, -6, -7, -8, -9, -10, -12(p70), -13, -15, -17, interferon (IFN)-γ, tumor necrosis factor (TNF)-α, Eotaxin, Granulocyte-macrophage colony-stimulating factor (GM-CSF), Granulocyte colony-stimulating factor (G-CSF), platelet-derived growth factor (PDGF)-BB, fibroblast growth factor-basic (FGF-basic), vascular endothelial GF (VEGF), IFN-γ-inducible-protein (IP)-10, macrophage chemotactic protein (MCP)-1, Regulated on Activation normal T cell expressed and activated (RANTES), macrophage inflammatory protein (MIP)-1α, and MIP-1β. Data were collected using Bio-plex manager software version 4 (Bio-Rad). A 5 Parameter Logistic (5 PL) regression formula was used to calculate sample concentrations from the standard curves. If the concentration of a cytokine fell outside of the limits of detection for a given sample, then that value was not used in statistical analysis. If any cytokine was undetectable in >40% of samples assayed, it was excluded from analyses beyond its initial description. Specimens from five participants were included across all plates (inter-plate controls). In addition, samples from five participants were duplicated on each set of plates (intra-plate controls) for quality control measures. Spearman's rank test was used to measure intra-assay and inter-assay correlation coefficients to determine assay reliability and reproducibility.

Cytokines were excluded from this analysis if the inter-assay correlation <0.8 (which included IL-12(p70)).

## Data analysis

Primer trimming was performed with cutadapt (1.16)[62]. Trimmed sequence data processing, classification, and amplicon sequence variant (ASVs) calling were performed using the DADA2 package (1.12.1)[63] within the R framework (3.6.1). ASVs were taxonomically classified using the Ribosomal Database Project's (RDP) Naïve Bayesian classifier[64]. Sequence classification was trained against an updated version of the Silva training set version 138[65], available at https://github.com/itsmisterbrown/updated_16S_dbs. Run-specific contamination filtering was performed using microfiltR (https://zenodo.org/badge/latestdoi/139792189), and samples with less than 5,000 filtered, annotated reads were discarded. Briefly, taxa represented by less than 0.01% of the total reads for a given sample were removed from that sample on a per-sample basis, and taxa that were present in less than 2.5% of samples or less than 0.015% of total reads across a sequencing run were discarded. For differential abundance testing, we further required that included ASVs were present in at least 15% of samples. Sequencing runs were merged using custom scripts available at https://zenodo.org/badge/latestdoi/200930263. The phyloseq (1.38.0)[66] and vegan (2.5–7)[67] packages were used for ecological analyses of bacterial communities. Inter-community distance was assessed using Bray-Curtis distance of relative abundance transformed abundance estimates or Euclidean distance of centered log ratio (CLR) transformed abundance estimates, as indicated. PERMANOVA was used to assess the significance of sample groupings by comparing a specified within-group centroid to the between-group centroid within the defined decomposed space (Bray-Curtis or Euclidean) using 1000 permutations. Partitioning around medoids (PAM) clustering was implemented using the cluster package (2.1.2)[68] and the number of clusters was selected using the gap statistic[69]. DESeq2 (1.34.0)[70] and ANCOM-BC (1.4.0)[71] were used to calculate the fold change in normalized ASV abundance across timepoints. All statistical models used in differential abundance testing in the primary (pre-post) analysis incorporated the effect of individual variation at baseline as a covariate, which also included the nested effect of study site, due to its significant effect on inter-community distance. Randomized contraceptive arm and study site were included as covariates in the secondary (case/control) analysis of bacterial differential abundance. For all differential abundance testing, the poscounts estimator was used in DESeq2 models and the ashr shrinkage estimator (2.2–54)[72] was used if models did not converge using the default approach. P-values from differential abundance testing via these approaches were adjusted using the method of Benjamini and Hochberg[41]. Bayesian directed acyclic graphs (DAGs) were generated on CLR transformed[73] bacterial abundance using the PC algorithm[74]. For the integrative analyses, bacterial inferred absolute concentrations were used for consistency with the measured absolute concentrations of cytokines. Both datasets were log₂ transformed and mean-centered prior to sparse principal component analysis.

**Statistics.** Alpha diversity estimates and cytokine concentrations were log-transformed prior to model fitting. For parameter estimation, error estimation, and model validation, the dataset was randomly and evenly partitioned into training and validation subsets. Statistical models were fit against the training subset and prediction accuracy (mean absolute error) was evaluated against the validation subset. Effect sizes were estimated from logistic regression models implemented as generalized linear models in R. Statistical tests used include permutational MANOVA (permanova) for comparisons among multiple (>2) groups in the beta-diversity analysis, ANCOVA models were used to evaluate the effect of contraceptive option on alpha diversity and Nugent scores while controlling for enrollment values, and Student's t tests or Wilcoxon Rank Sum tests for comparisons among two groups. McNemar's

and chi-squared tests were used for assessing data in contingency tables. Regardless of the test used, significance was evaluated using an α value cutoff of 0.05. *P*-value adjustment for multiple comparisons was performed according to Benjamini and Yekutieli[42], unless otherwise noted. When parametric tests were used, assumptions concerning the homogeneity of variances (Bartlett) and normality of data distribution (Shapiro) were evaluated with the appropriate test. Boxplots display the median and interquartile range.

**Bacterial growth curves.** Growth curves were generated for *Lactobacillus crispatus* (MV-1A-US, BEI Resources), *Streptococcus agalactiae* (GBS; COH1)[75], and *Escherichia coli* (DH10B, Thermo Fisher Scientific). *L. crispatus* was obtained through BEI Resources, NIAID, NIH as part of the Human Microbiome Project: Lactobacillus crispatus, Strain MV-1A-US, HM-637. GBS COH1 was donated by Laksmi Rajagopal at Seattle Children's Research Institute. Cultures were started from stocks frozen at −80 °C by transferring to agarose plates (MRS for *L. crispatus*, TSA for *GBS*, TSA for *E. coli*) and incubating overnight at 37 °C. Single colonies from each plate were transferred to liquid cultures (MRS for *L. crispatus*, TSB for GBS, LB for *E. coli*) and incubated overnight at 37 °C for *L. crispatus* and *E. coli* and 30 °C for *GBS*. All cultures were grown aerobically. Dilutions of Copper chloride (Sigma) solution corresponding to concentrations of $Cu^{2+}$ from $9e^{-4}$ to $9e^{-10}$ g/mL were prepared as previously described[49]. The optical density ($OD_{600}$) of each bacterial culture was measured after overnight incubation and each suspension was diluted to $OD_{600}$ 0.1. Diluted bacterial suspensions were mixed with the appropriate culture media and copper chloride dilution to achieve the designated concentrations for growth measurements in a total of 200 μL in each well of a 96-well plate. Bacterial growth was assayed across three independent culture wells for each bacterial taxa at each copper concentration (biological triplicates). Each experiment was repeated twice from fresh cultures.

**Quantification of total bacterial abundance and taxon-specific inferred concentrations by qPCR**
Total bacterial abundance (16S rRNA copies/μL) and the inferred concentrations of specific bacterial taxa (16S rRNA copies/μL) were performed as previously described[76]. Standards from 1e7 to 10 copies and a no-template control were run in triplicate across all plates. Samples, primers, and probes were used with SsoAdvanced Universal Probes Supermix (Bio-Rad) or Platinum Quantitative PCR SuperMix-UDG w/ROX (Invitrogen) kits and run on a StepOnePlus Real-Time PCR System (Applied Biosystems). All samples were assayed in duplicate and were discarded if the Ct standard deviation was flagged as high or the Ct values were outside of the standard curve. Four samples had less than 100 total 16S rRNA copies and were not used in downstream analyses. Inferred concentrations of individual bacterial taxa were determined using Eq. 1:

$$IC_x = RAb_x \times Ab_{total} \tag{1}$$

Where $IC_x$ is the inferred concentration of taxon $x$, $RAb_x$ is the relative abundance of taxon $x$, and $Ab_{total}$ is the total bacterial abundance.

**Reporting summary**
Further information on research design is available in the Nature Portfolio Reporting Summary linked to this article.

## Data availability
The processed 16S count data and cytokine data generated in this study have been deposited in the Dryad database under accession code https://doi.org/10.5061/dryad.3n5tb2rmv. The raw nucleotide sequence data have been deposited in the National Center for Biotechnology Information Sequence Read Archive under accession PRJNA918342.

## Code availability
All custom functions and the R code necessary to reproduce the analyses performed here are available at https://zenodo.org/badge/latestdoi/242246946.

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

## Acknowledgements

We thank all the women who participated in this study for their devotion to the study and the time they committed to research procedures. This work and the Evidence for Contraceptive Options and HIV Outcomes (ECHO) Study were made possible by the combined generous support of the Bill & Melinda Gates Foundation (Grant OPP1032115), the American people through the United States Agency for International Development (Grant AID-OAA-A-15–00045), the Swedish International Development Cooperation Agency (Grant 2017/762965–0), the South Africa Medical Research Council, and the United Nations Population Fund. Contraceptive supplies were donated by the Government of South Africa and US Agency for International Development. Funding to support this ancillary study of biological mechanisms was from the US National Institute of Child Health and Human Development under grant numbers 5R01HD089831 (R.H., H.B.J.). B.P.B. was also supported by F32HD102290 and K99HD106861. The content is the sole responsibility of the authors and does not necessarily represent the official views of the study funders.

## Author contributions

S.E.B., A.B., J.S.P., R.H., and H.B.J. designed the microbiota sub-study to the ECHO Trial. B.P.B., C.F., R.F.T., S.Z.J., R.B., S.D., D.D.N., C.F., M.G., and A.H. performed the experiments. M.O., G.N., T.P.P., C.W.S., and H.K. provided logistical support, data management and oversaw all data collection. B.P.B. wrote the first draft of the manuscript. All authors reviewed and approved the final manuscript.

## Competing interests
The authors declare no competing interests.

## Additional information

[1]Seattle Children's Research Institute, Seattle, USA. [2]University of Washington, Seattle, USA. [3]Institute of Infectious Diseases and Molecular Medicine, University of Cape Town, Cape Town, South Africa. [4]The Medical Research Centre, Institute of Medical Research and Medicinal Plant Studies (IMPM), Ministry of Scientific Research and Innovation, Yaoundé, Cameroon. [5]Kenya Medical Research Institute, Kisumu, Kenya. [6]Desmond Tutu HIV Centre, Cape Town, South Africa. [7]Wits Reproductive Health and HIV Institute, Johannesburg, South Africa. [8]Gilead Sciences, Inc, Seattle, USA. [9]Yerkes National Primate Research Center, Atlanta, USA. [10]Emory University, Atlanta, USA. [11]Center for Global Health and Diseases, Case Western Reserve University, Cleveland, USA. [12]University of Manitoba, Winnipeg, Canada. [13]Karolinska Institute, Stockholm, Sweden. [14]National Health Laboratory Service, Cape Town, South Africa. [15]University of Alabama, Birmingham, USA. ✉e-mail: bryan.brown@seattlechildrens.org; hbjaspan@gmail.com

