## [Peer Review File · Nature Communications]

REVIEWER COMMENTS

Reviewer #2 (Remarks to the Author):

Overall this updated version of the manuscript is substantially improved. The authors have clarified some of the issues I previously raised and where possible, have addressed most of my concerns. The new bacterial load data provides important new insight into how the microbiome is dynamically changing in response to the Cu-IUD and helps differentiate the work presented here from previous studies that have investigated aspects of vaginal microbiota response to Cu-IUD. I still feel that the lack of a genuine control/placebo arm in the study design is a major limitation that should be more openly and frankly discussed. However, the work is of sufficient quality and interest to the community that I am happy to recommend it now be accepted for publication.

Reviewer #3 (Remarks to the Author):

This paper reports the primary analysis of data from a substudy of a RCT comparing three contraceptives. My main concerns are around the sample selection and the statistical methods being used to compare the groups (see below), but I also consider the writing to lack clarity, such that I found it frequently difficult to understand what had been done.

Major points

1. Sample derivation:

a. Fig S1 says “After completion of follow-up, 20 participants per arm were randomly selected from each site”. This would suggest sample sizes of 60 per arm, but they are actually larger than this.

Why?

b. Methods says differently, “20-30 women per contraceptive arm per site” – please resolve the contradiction, and if the latter is correct, please explain how a number was chosen between 20 and 30.

c. Methods adds that certain women were randomly selected within the mucosal substudy – this seems to represent a second stage of random selection and needs to be explained and justified.

d. Methods says “we consecutively enrolled all eligible women concurrent to their enrolment in the primary trial, or thereafter if already enrolled” – why might a woman have already been enrolled? perhaps because enrolment to this substudy started late? So when did it start and why?

2. Main comparisons (l132-145):

a. Key point: The authors use before-after within arms which is not a valid approach. They should use a comparison of arms at 1 and 6 months adjusted for baseline (e.g. ANCOVA) instead. (Consider use of ordered logistic regression to perform the testing with adjustment.)

b. Statistical tests on baseline data between randomised groups (as done in Figure 1) are deprecated (see e.g. <https://onlinelibrary.wiley.com/doi/10.1002/sim.4780131703>).

c. Nugent score is an integer in the range 0-10, so the plots used in Fig 1A are inappropriate. Histograms can be used instead.

d. Figure 1B and 1C show the same data which is not desirable.

3. Results on “Contraceptive use explains shifts in inter-community distance” (l147-174) are unclear:

a. Key point: The term “inter-community distance” is used frequently but not defined. I assume it refers to “distance in bacterial communities between some groups”, but it is not clear what the groups are. Initially I thought the groups were the randomised arms, but then l161-2 “Randomized contraceptive had a significant effect on inter-community distance” becomes meaningless. Please explain.

b. The Methods section does not explain how PERMANOVA is linked to Bray-Curtis. I assume the null hypothesis is that the Bray-Curtis distance is one. Please state this.

c. l150-1 takes a non-significant test as proving the null hypothesis is true (here, that the arms were comparable). This is faulty logic & should be corrected (even though the conclusions is probably true).

d. I believe the main point of this paragraph is contained in lines 161-6 and should be expressed as “Contraceptive use causes shifts in bacterial community profiles”.

4. To me, giving results in subheadings is unscientific. Subheadings should relate to questions, not answers.

Some minor points

5. Table 1: too many decimal places. Integer values should be reported as integers (e.g. gravidity). Consider reporting age and BMI to 0 or 1 decimal places. Add % to percentages for clarity e.g. “14 (23%)”

6. p4, “The prevalence of Chlamydia trachomatis differed by contraceptive arm ... though this did not have a significant effect on bacterial community alpha- or beta-diversity”: please explain the latter statement.

7. l137, “Figure 1A-B” – should be 1B.

8. l151-3, “There was also no significant difference between alpha diversity and participants who tested positive for C. trachomatis or N. gonorrhoeae at enrolment” – you can’t compare alpha diversity, which is a number, with participants, who are people. I don’t know what you mean. Please re-write.

9. l541-2, “Linear models between alpha diversity measures and cytokine concentrations were also log transformed prior to model generation” – you can’t transform a linear model. Please state which variables were log-transformed.

REVIEWER COMMENTS

Reviewer #2 (Remarks to the Author):

Overall this updated version of the manuscript is substantially improved. The authors have clarified some of the issues i previously raised and where possible, have addressed most of my concerns. The new bacterial load data provides important new insight into how the microbiome is dynamically changing in response to the Cu-IUD and helps differentiate the work presented here from previous studies that have investigated aspects of vaginal microbiota response to Cu-IUD. I still feel that the lack of a genuine control/placebo arm in the study design is a major limitation that should be more openly and frankly discussed. However, the work is of sufficient quality and interest to the community that I am happy to recommend it now be accepted for publication.

>We are happy to hear that the reviewer feels that we have addressed their concerns and substantially improved the manuscript. We agree that the lack of a placebo arm is a limitation of the study that, unfortunately, we were not able to alter. We have expanded our discussion of this limitation considering this recommendation (I422-427).

Reviewer #3 (Remarks to the Author):

This paper reports the primary analysis of data from a substudy of a RCT comparing three contraceptives. My main concerns are around the sample selection and the statistical methods being used to compare the groups (see below), but I also consider the writing to lack clarity, such that I found it frequently difficult to understand what had been done.

Major points

1. Sample derivation:

a. Fig S1 says “After completion of follow-up, 20 participants per arm were randomly selected from each site”. This would suggest sample sizes of 60 per arm, but they are actually larger than this. Why?

> We appreciate the attention to detail paid by the reviewer here, and regret the lack of consistency. Higher numbers were selected from the Emavundleni site (Cape Town) due to the availability of matching flow cytometer at that site for analysis of matching cervical T cell data (Bunjun et al, CID in Press) (I646-648). This has been clarified in the methods and we have updated Figure S1 accordingly.

b. Methods says differently, “20-30 women per contraceptive arm per site” – please resolve the contradiction, and if the latter is correct, please explain how a number was chosen between 20 and 30.

>Please see response to a. above.

c. Methods adds that certain women were randomly selected within the mucosal substudy – this seems to represent a second stage of random selection and needs to be explained and justified.

>See response to d. below

d. Methods says “we consecutively enrolled all eligible women concurrent to their enrolment in the primary trial, or thereafter if already enrolled” – why might a woman have already been enrolled? perhaps because enrolment to this substudy started late? So when did it start and why?

>We apologize for the confusion caused. We have tried to clarify within the text (l646-648). For additional clarity we explain here;

A participant may have already been enrolled into the parent trial because enrollment for the parent trial commenced prior to the substudy. The substudy started enrolling participants about 9 months into the start of the main trial due to delayed funding. Therefore, some women enrolled at baseline, and some opted to enroll at later visits. The latter participants were not included in the pre-post contraception analysis, but only in the case-control analysis of we had a sample taken prior to the seroconversion visit.

There was only one layer of random selection for the pre-post analysis which was randomly selecting 20 - 30 women per site, except for the Cape Town site as explained above. For the seroconverter analysis, we randomly selected controls matched by age category for the seroconverter cases as described in l651 - 652.

2. Main comparisons (l132-145):

a. Key point: The authors use before-after within arms which is not a valid approach. They should use a comparison of arms at 1 and 6 months adjusted for baseline (e.g. ANCOVA) instead. (Consider use of ordered logistic regression to perform the testing with adjustment.)

>We thank the reviewer for this suggestion and have reanalyzed our data accordingly. Using an ANCOVA (adjusted using Tukey’s correction), as suggested, we find that Shannon diversity is significantly higher among participants randomized to Cu-IUD than DMPA-IM ($P= 0.02$) after six months of use and when adjusting for baseline diversity (l149). However, we feel that comparisons of bacterial diversity and Nugent score between enrollment and at each time point also provide valuable information not yielded from cross-sectional comparisons, and this approach has been employed in several other studies evaluating BV frequency relative to baseline[1], and bacterial diversity and inflammation relative to baseline [2-4]. Additionally, a previous reviewer had requested this analysis and therefore we have elected to retain it.

b. Statistical tests on baseline data between randomised groups (as done in Figure 1) are deprecated (see e.g. <https://onlinelibrary.wiley.com/doi/10.1002/sim.4780131703>).

>Thank you for bringing this to our attention. We appreciate that this analysis may be superfluous but we had added this in response to the first round of reviews.

c. Nugent score is an integer in the range 0-10, so the plots used in Fig 1A are inappropriate. Histograms can be used instead.

> As the reviewer has commented, Nugent scores are an integer, which can be displayed as boxplots, (See [5]). If we display this instead as histograms, we would need to expand this subfigure to nine facets (as opposed to three in the previous version) as shown below, but can do so if the reviewer/ editor insists.

d. Figure 1B and 1C show the same data which is not desirable.

>We have moved Figure 1C to supplemental information.

3. Results on “Contraceptive use explains shifts in inter-community distance” (I147-174) are unclear:

a. Key point: The term “inter-community distance” is used frequently but not defined. I assume it refers to “distance in bacterial communities between some groups”, but it is not clear what the groups are. Initially I thought the groups were the randomised arms, but then I161-2 “Randomized contraceptive had a significant effect on inter-community distance” becomes meaningless. Please explain.

>We thank the reviewer for pointing out the lack of clarity here. They are correct in assuming that inter-community distance refers to distance between bacterial communities in a sample. We use a permutational ANOVA to assess significance of grouping variables (eg. randomized arm, study site, etc.), by comparing the distance within a grouping to the overall centroid, which is assessed via permutation. We have elaborated on this and clarified our phrasing in lines 157 – 159.

b. The Methods section does not explain how PERMANOVA is linked to Bray-Curtis. I assume the null hypothesis is that the Bray-Curtis distance is one. Please state this.

>Similar to our response above, we appreciate the reviewer noting this and have clarified in the methods section as requested (I157-159 and I733-735). H_0 for a permanova is that the centroids of the groups are equivalent in the defined space, which in our case is the decomposed space derived from Bray-Curtis distance.

c. I150-1 takes a non-significant test as proving the null hypothesis is true (here, that the arms were comparable). This is faulty logic & should be corrected (even though the conclusions is probably true).

>We have adjusted our phrasing to more accurately communicate that we did not find a significant difference in groups at baseline, which lead us to believe that there was not an underlying community structure biasing our results (l159-162).

d. I believe the main point of this paragraph is contained in lines 161-6 and should be expressed as “Contraceptive use causes shifts in bacterial community profiles”.

> We agree and have adjusted accordingly.

4. To me, giving results in subheadings is unscientific. Subheadings should relate to questions, not answers.

>We defer to the editorial staff for their preference.

Some minor points

5. Table 1: too many decimal places. Integer values should be reported as integers (e.g. gravidity). Consider reporting age and BMI to 0 or 1 decimal places. Add % to percentages for clarity e.g. “14 (23%)”

>We agree and have adjusted accordingly.

6. p4, “The prevalence of Chlamydia trachomatis differed by contraceptive arm ... though this did not have a significant effect on bacterial community alpha- or beta-diversity”: please explain the latter statement.

>We thank the reviewer for noticing this. There was supposed to a reference to Figure S3, which shows the analysis of Ct and Ng on alpha and beta diversity. We have added this reference.

7. l137, “Figure 1A-B” – should be 1B.

>Adjusted accordingly

8. l151-3, “There was also no significant difference between alpha diversity and participants who tested positive for C. trachomatis or N. gonorrhoeae at enrolment” – you can’t compare alpha diversity, which is a number, with participants, who are people. I don’t know what you mean. Please re-write.

>Thank you, we have reworded for clarity (l152-154).

9. l541-2, “Linear models between alpha diversity measures and cytokine concentrations were also log transformed prior to model generation” – you can’t transform a linear model. Please state which variables were log-transformed.

>As above, we regret the confusing phrasing and have adjusted for clarity (l742).

1. Peebles, K., et al., *Elevated Risk of Bacterial Vaginosis among Users of the Copper Intrauterine Device: A Prospective Longitudinal Cohort Study*. Clinical Infectious Diseases, 2020.

2. Balle, C., et al., *Hormonal contraception alters vaginal microbiota and cytokines in South African adolescents in a randomized trial*. Nat Commun, 2020. **11**(1): p. 5578.
3. Paramsothy, S., et al., *Multidonor intensive faecal microbiota transplantation for active ulcerative colitis: a randomised placebo-controlled trial*. Lancet, 2017. **389**(10075): p. 1218-1228.
4. Tanko, R.F., et al., *The Effect Of Contraception On Genital Cytokines In Women Randomized To Copper Intrauterine Device, Intramuscular Depot Medroxyprogesterone Acetate Or Levonorgestrel Implant*. The Journal of Infectious Diseases, 2022.
5. Nunn, K.L., et al., *Vaginal glycogen, not estradiol, is associated with vaginal bacterial community composition in black adolescent women*. Journal of Adolescent Health, 2019. **65**(1): p. 130-138.

REVIEWER COMMENTS

Reviewer #3 (Remarks to the Author):

My main two points have not been adequately addressed.

Major points

1. Sample derivation

The authors are now consistent in their description of the sample size. They say “20 - 30 women per contraceptive arm per site ... were randomly selected ...”. But they still do not say how the figure between 20 and 30 was selected. This should be reported clearly.

I do not believe that the authors initially selected 20 women, ran the analyses, decided they didn't like the results, and selected a few more women. But that is consistent with what the authors describe.

2. Main comparisons: The authors use before-after within arms which is not a valid approach.

The authors have adopted the correct approach (comparison between randomised groups) for one outcome (Shannon diversity) but not for the two other outcomes reported here. As a result, this paragraph still largely consists of incorrect before-after comparisons with just two correct comparisons of randomised groups (labelled as “cross-sectional”).

The authors give as their reason for performing the incorrect analysis that it was requested by a previous reviewer. It is for the editor to determine whether that previous reviewer's advice should take precedence over this statistical reviewer's advice on statistical matters.

It is worth noting that the authors do not appear to have pre-specified a statistical analysis plan and are therefore at risk of charges of data-dredging.

Reviewer #3 (Remarks to the Author):

My main two points have not been adequately addressed.

Major points

1. Sample derivation

The authors are now consistent in their description of the sample size. They say “20 - 30 women per contraceptive arm per site ... were randomly selected ...”. But they still do not say how the figure between 20 and 30 was selected. This should be reported clearly.

I do not believe that the authors initially selected 20 women, ran the analyses, decided they didn't like the results, and selected a few more women. But that is consistent with what the authors describe.

> We regret the lack of transparency that remained in our previous revision. Twenty participants per randomized arm were selected from the sites in Johannesburg, South Africa, and Kisumu, Kenya. At the Cape Town site, 80 consecutively enrolled participants, regardless of arm, were included in a separate *ex vivo* analysis of cervical T cells, which was being performed at this study site due to the ability to perform multiparameter flow cytometry on fresh specimens (Bunjun et al. *in Press*). Therefore, we elected to include all of these women, rather than a random subset. The manuscript describing this work has been accepted for publication in *Clinical Infectious Diseases* but has not been published yet, as such we are happy to provide a copy of the accepted manuscript if desired.

Thus, the process that led to our final sample sizes was as follows: 60 participants were randomly selected from the sites in Johannesburg, South Africa, and Kisumu, Kenya (n=20 per randomized arm per site). From the Cape Town, South Africa site, 80 participants were selected based on consecutive enrollment (27 in the Cu-IUD arm, 28 in the DMPA-IM arm and 25 in the LNG implant arm) to correspond with a separate analysis of cervical T cells (Bunjun et al. *in Press*). Additionally, participants from the case-control group were added if samples from all visits (enrollment, one month, and six months) were available and they were not already in the pre-post group. Finally, samples were removed if they did not pass quality control (see supplemental Table 3A).

In an effort to increase the clarity around our sample sizes and selection, we have updated Table S3 to include the sample counts for the primary (pre-post) analysis, the sample counts for the secondary (case-control) analysis (Table S3B), and the sample counts for participants that overlapped and were included in both analyses (Table S3C). Changes in the manuscript text are reflected in I530-552.

2. Main comparisons: The authors use before-after within arms which is not a valid approach.

The authors have adopted the correct approach (comparison between randomised groups) for one outcome (Shannon diversity) but not for the two other outcomes reported here. As a result, this paragraph still largely consists of incorrect before-after comparisons with just two correct comparisons of randomised groups (labelled as “cross-sectional”).

The authors give as their reason for performing the incorrect analysis that it was requested by a

previous reviewer. It is for the editor to determine whether that previous reviewer's advice should take precedence over this statistical reviewer's advice on statistical matters.

> We have redone the analyses as recommended by reviewer 3, with all statistical tests now reflecting comparisons between randomised groups with adjustment for baseline measures. It was an unfortunate omission in the previous revision that we did not adjust the analysis of Nugent score data to account for baseline values and we have corrected that herein. In our previous revision, our beta-diversity analyses (the permanova tests) did include an adjustment for baseline diversity, but this may not have been clear. In response, we have rewritten this section to provide additional clarity regarding our adjustment for baseline diversity in our beta-diversity analyses. We have moved longitudinal analyses into the supplementary figures and we have rewritten the text (lines 135-170; lines 341-344) to prioritize the recommended analyses. Figure 1 now exclusively displays cross-sectional results that have been adjusted for baseline values across randomized arms, as recommended.

It is worth noting that the authors do not appear to have pre-specified a statistical analysis plan and are therefore at risk of charges of data-dredging.

> We appreciate this advice and have now included our concept sheet outlining sample types, analysis plan, and requested metadata from the parent trial consortium that was developed prior to randomization unblinding. This document describes the statistical analyses, data transformations, and hypothesis tests that were originally proposed for use on these data, demonstrating that data dredging truly did not occur as this analysis was a primary focus of our grant proposal and the plans for using these data.

I have now reviewed all the comments made by Reviewer #3 and I agree with their comments. Indeed, their comments were very thorough. I think authors' responses are adequate. However, I have a few additional comments on the revision. I am referring to the pdf file "358486_2_merged_1659988837". Authors sentences are in italics and my comments are in standard text following the authors' sentences

1. *Authors: Page 6 - "We used DESeq2 to identify taxa that were significantly differentially abundant after 6 months ..."*

Comment: If I understood correctly, the authors have paired data within each arm. If I am not mistaken, the DESeq2 is designed for independent samples and not paired samples. Please clarify how the normalization within DESeq2 was performed and the data were subsequently analyzed as these are paired data. Secondly, it is well established in the literature that DESeq2 is prone to inflated FDR. Hence the authors should consider other methods such as ANCOM-BC, LOCOM and LiNDA.

2. *Authors: Page 7 - "Cu-IUD use results in significant increases in total bacterial load and abundance of several BV-associated taxa"*

Comment: In the middle of the paragraph the authors state that they used Wilcoxon's test for performing differential abundance analysis. I am confused why they did that when earlier on page 6 they stated that they used DESeq2. Again, Wilcoxon's test is also prone to inflated FDR. What is worse, from the description in the paragraph it does not appear that the authors have controlled for multiple testing. Some of the p-values, without correcting for multiple testing are barely significant (e.g. *A. vaginae* $P = 0.032$, and others). Thus, I am not convinced about the findings of the differential abundance analysis.

3. *Authors: Page 8 - "Integrative compositional analyses reveal alterations in microbiota and inflammation between cases and controls"*

Comment: Not clear what method of differential abundance analysis was used.

4. *Authors: Page 8 - "Sparse PC analysis"*

Comment: I do not understand the rationale for using SPC4 or 2? Why selected these specific PCs. Seems arbitrary/ad-hoc.

We thank Reviewer 4 for their thoughtful comments (listed in black) and have responded to each point below in orange text.

Comment: If I understood correctly, the authors have paired data within each arm. If I am not mistaken, the DESeq2 is designed for independent samples and not paired samples. Please clarify how the normalization within DESeq2 was performed and the data were subsequently analyzed as these are paired data. Secondly, it is well established in the literature that DESeq2 is prone to inflated FDR. Hence the authors should consider other methods such as ANCOM-BC, LOCOM and LiNDA.

> We appreciate the value of additional analysis and a conservative interpretation of our results and have thus performed additional differential abundance analyses and a reanalysis of our DESeq2 data with a more conservative approach. We agree with the editors that there is no consensus method for differential abundance analysis in 16S studies and have thus chosen to include the use of ANCOM-BC as it is known to be one of the more conservative tools for such analyses and uses compositional data analytical techniques. We have replaced the differential abundance analysis of our pre-post study with results obtained from ANCOM-BC (Figure 2B) and have updated the text accordingly (lines 210 - 231). Additionally, we performed a reanalysis of our data with DESeq2 using a much more conservative abundance estimator (the 'poscounts' approach) that was designed for 16S data with high levels of sparsity (Figure S5). While this was conceptually similar to the normalization procedure that we had used previously (a geometric mean estimator), it assumes that zero counts are truly absent and penalizes them more harshly during differential abundance analysis. For both tools, these analyses were performed only on paired samples and individual variation was specifically accounted for as a covariate in the model we used; p-value correction was performed using the Benjamini-Hochberg procedure, which is a powerful correction for controlling the FDR. Generally, there was good concordance between the methods with consistent taxa detected by both tools though, surprisingly, ANCOM-BC detected a greater number of differentially abundant taxa than DESeq2.

Comment: In the middle of the paragraph the authors state that they used Wilcoxon's test for performing differential abundance analysis. I am confused why they did that when earlier on page 6 they stated that they used DESeq2. Again, Wilcoxon's test is also prone to inflated FDR. What is worse, from the description in the paragraph it does not appear that the authors have controlled for multiple testing. Some of the p-values, without correcting for multiple testing are barely significant (e.g. A. vaginae P = 0.032, and others). Thus, I am not convinced about the findings of the differential abundance analysis.

>We used a Wilcoxon test to compare the inferred absolute concentrations of all taxa with greater than 30% prevalence between baseline and 6 months for each randomized arm. Because these data were not compositional (as was the case for the data used in the DESeq2 and ANCOM-BC analysis) and reflected absolute values, a Wilcoxon test was appropriate and the reported p-values were corrected using the Benjamini-Yekutieli method, which is a very powerful, and notably conservative, correction method that

extends the Benjamini-Hochberg correction to a wider variety of datasets and problems. Since this was not noted by the reviewer, we have more clearly specified this in the text (lines 251-252).

Comment: Not clear what method of differential abundance analysis was used.

> We have reframed this section to focus more broadly on the integrative bacterial-cytokine analysis of the case/control dataset. We have moved the differential abundance testing results to the end of the section and clarified this and the results by each test in the text in lines 334-339.

Comment: I do not understand the rationale for using SPC4 or 2? Why selected these specific PCs. Seems arbitrary/ad-hoc.

> We regret the lack of clarity here. We decomposed our data into 5 sparse principal components (SPCs) as this number explained greater than 60% of the variation in the case/control dataset. We further examined SPC4 as this was the component that significantly demarcated cases from controls (Figure 5A). SPC2 was not discussed further as that was a typographical error from an earlier revision that we have since corrected.